# Susceptibility Loci for Type 2 Diabetes in the Ethnically Endogamous Indian Sindhi Population: A Pooled Blood Genome-Wide Association Study

**DOI:** 10.3390/genes13081298

**Published:** 2022-07-22

**Authors:** Kanchan V. Pipal, Manju Mamtani, Ashwini A. Patel, Sujeet G. Jaiswal, Manisha T. Jaisinghani, Hemant Kulkarni

**Affiliations:** 1Lata Medical Research Foundation, Nagpur 440002, India; pipalkanchan17@gmail.com (K.V.P.); manju.mamtani@mnhresearch.com (M.M.); ashwinipatel1993@gmail.com (A.A.P.); sujeetgjaiswal@gmail.com (S.G.J.); manishajaisinghani1001@gmail.com (M.T.J.); 2M&H Research, LLC, San Antonio, TX 78249, USA

**Keywords:** type 2 diabetes, genome-wide association study, ethnicity

## Abstract

Type 2 diabetes (T2D) is a complex metabolic derangement that has a strong genetic basis. There is substantial population-specificity in the association of genetic variants with T2D. The Indian urban Sindhi population is at a high risk of T2D. The genetic basis of T2D in this population is unknown. We interrogated 28 pooled whole blood genomes of 1402 participants from the Diabetes In Sindhi Families In Nagpur (DISFIN) study using Illumina’s Global Screening Array. From a total of 608,550 biallelic variants, 140 were significantly associated with T2D after adjusting for comorbidities, batch effects, pooling error, kinship status and pooling variation in a random effects multivariable logistic regression framework. Of the 102 well-characterized genes that these variants mapped onto, 70 genes have been previously reported to be associated with T2D to varying degrees with known functional relevance. Excluding open reading frames, intergenic non-coding elements and pseudogenes, our study identified 22 novel candidate genes in the Sindhi population studied. Our study thus points to the potential, interesting candidate genes associated with T2D in an ethnically endogamous population. These candidate genes need to be fully investigated in future studies.

## 1. Introduction

It is well established that type 2 diabetes (T2D) has a strong genetic basis. Studies around the world report that T2D as a trait has a heritability ranging from 35–80% indicating that a large proportion of the variability in T2D can be explained by genetics [1,2,3,4,5]. Genome-wide association studies have identified several potential single nucleotide polymorphisms (SNP) that significantly contribute to the genetic basis of T2D [6]. Still, it is recognized that the strength of association of these SNPs with T2D can vary across world populations [7,8,9,10]. In that vein, a search for population-specific genetic determinants of T2D still continues.

The Indian Sindhi ethnically endogamous group has historically encountered the challenges of stress and migration. The partition of India and Pakistan in 1947 led to an exodus of Hindu Sindhis residing in the Sindh province (now in Pakistan) to various parts across India. This exodus remains a critical root of the chronic stresses experienced by generations of Indian Sindhis [2,11]. Combined with these stressors, the energy-rich diet and relative lack of physical activity place this population at an enhanced risk of T2D. Despite these well-known environmental determinants of T2D, in the first study of extended and complex Sindhi pedigrees, we observed that the heritability of T2D was 35% [2]. In the era of personalized medicine, this information is important since individual susceptibility to T2D and the individual’s response to T2D treatment can both be determined, in part, by the genetic disposition [12]. To that end, a genome-level interrogation of the potential markers of T2D in the Indian Sindhi population is needed.

We conducted a pooled blood genome-wide association study of T2D in the pedigrees of Indian Sindhi families enrolled in the Diabetes In Sindhi Families In Nagpur (DISFIN) study [2]. This population has a substantial coexistence of metabolic comorbidities such as hypertension, dyslipidemia, general obesity and central obesity. We used innovative statistical approaches to account for these comorbidities and—constrained by the project costs—conducted a pooled whole blood genome-wide association study (GWAS). Here, we report the results of our study that identified interesting leads into the genetic basis of T2D in the Indian Sindhi families.

## 2. Materials and Methods

### 2.1. Study Participants

The clinical and genetic data for the present study came from the participants in the DISFIN study. Details of the enrollment protocol, inclusion and exclusion criteria and overall design of the study have been described elsewhere [2]. Briefly, we enrolled endogamous Sindhi families to construct family pedigrees. The pedigrees ranged from simple nuclear ones to complex and extended three generation families. Consenting participants with at least one known patient of T2D in the family were included in the study. Other inclusion criteria were a resident of the study area (Jaripatka, Mecosabag and Khamla areas of Nagpur where the Sindhi ethnic population mostly resides); self-reported Sindhi ethnicity and age ≥20 years. Pregnant or lactating women and patients with type 1 diabetes (known or suggested by serum C peptide) were excluded. After investigator-administered, semi-structured interviews and clinical examination, a trained phlebotomist collected blood samples for laboratory assays. Part of the sampled venous blood was stored at −80 °C to conduct genetic investigations later. Participant enrollment and blood sample collection took place between 1 March 2016 and 28 February 2017.

### 2.2. Definitions of Metabolic Conditions

The primary goal of the study was to conduct a genome-wide interrogation in the context of T2. Since metabolic comorbidities such as obesity, hypertension and dyslipidemia commonly coexist with T2D, we also investigated the study participants with respect to these conditions. T2D was defined [13] as self-reported diabetes OR currently on anti-diabetics OR fasting plasma glucose ≥ 126 mg/dL OR random blood glucose ≥ 200 mg/dL OR HbA1c concentration ≥ 6.5%. Hypertension was defined [14] as self-reported hypertension OR currently on anti-hypertensives OR systolic blood pressure ≥ 130 mmHg OR diastolic blood pressure ≥ 85 mmHg. Central obesity was defined based on cutoffs for Indian population [15] as a waist circumference ≥ 90 cm for males and ≥85 cm for females. Dyslipidemia was defined [16] as presence of any of the following: serum triglycerides ≥ 150 mg/dL and serum high density lipoproteins < 40 mg/dL (for males) or <50 mg/dL (for females).

### 2.3. Pool Definitions for GWAS

We conducted a pooled whole blood genome-wide association study. This technique is now well established as an acceptably accurate and inexpensive alternative to individual genome-wide genotyping [17]. To account for the potential contribution of metabolic comorbidities to genome-wide associations, we constructed whole blood pools based on a combination of the presence of T2D, central obesity, hypertension and dyslipidemia. Using these four binary traits we first generated a total of 16 potential combinations. We combined those categories that had a frequency < 1% with preceding categories. This strategy resulted in a total of 14 adequately represented pools that were based on the metabolic comorbidity profile. To ensure repeatability, we ran the genotyping analyses on duplicate pools. Therefore, we had a total of 28 whole blood pools that were used for genotyping analyses.

### 2.4. Whole Blood Pool Construction and Genotyping

We aimed to collect at least 3 µg of DNA for each pool from the K-EDTA stored blood samples were thawed gradually to room temperature in a vibration free environment. From these samples, 100 µL of thawed blood was pipetted per sample and added to specific pools defined by the comorbidity profile. From these pooled blood samples, two aliquots of 2 mL each were prepared, thus resulting in 28 pooled samples for GWAS. Whole blood samples were transported to the genotyping laboratory within 6 h on dry ice. All genotyping was carried out at the Genetics Laboratory, MedGenome Labs Ltd. (https://diagnostics.medgenome.com/, accessed on 30 June 2022), Bangaluru, India. Genotyping of the pool samples was conducted using Illumina’s Infinium Global Screening Array (GSA) platform from harvested genomic DNA (Qiagen, DNeasy Blood and Tissue Kit). A total of 200 ng of genomic DNA was amplified and incubated on day 1. This was followed next day with fragment amplification, precipitation, resuspension, bead chip preparation and hybridization of the sample on the bead chip. Stained samples with red and green channel tags on the bead chip were then imaged and subjected to an auto calling algorithm to generate the B allele frequencies in the pools.

### 2.5. Statistical Analyses

For each included variant on the GSA array, we had 28 estimations of the B allele frequencies. We accounted for the potential of confounded associations ascribable to the presence of comorbidities, batch effects, intra-replicate correlation, within-pool degree of kinship and the random effects across pools. For this we used the mixed effects logistic regression format to estimate the T statistic from a Wald test to test the significance of association of a given variant with the risk of T2D. Specifically, we used the following regression model to estimate the strength of association:logit(T2D) = β_0_+ β_s_ BAF + β_1_ HT + β_2_ COB + β_3_ DYL + β_b_ BAT + β_r_ REP + β_k_ PHI + RE(POOL)
where, T2D is an indicator variable for presence of T2D; BAF is a continuous variable indicating the B allele frequency; HT, COB and DYL represent the concomitant presence of hypertension, central obesity and dyslipidemia, respectively; BAT is an indicator variable for the chip identifier; REP represents the replicate ID; PHI is the within pool degree of kinship and RE(POOL) represents the random effects across the study pools. The regression coefficients in the equation were used to quantify the differential B allele frequency (β_s_), influence of comorbidities (β_1_–β_3_), batch effect (β_b_), pooling error (β_r_) and kinship effect (β_k_). All the models used a weighted approach based on pool frequency weights. The rigid Bonferroni correction was used to account for multiple testing and the global type I error rate was thus adjusted to 8.216 × 10^−8^. These analyses were conducted using dedicated scripts in R. Manhattan and QQ plots were generated using the qqman library [18] in R. Pooling error was estimated as described by MacGregor et al. [19].

### 2.6. Functional Relevance of Strongly Associated Variants and Genes

To understand the detected associations from a functional perspective, we used three approaches. First, all the variants and genes were annotated using the DisGeNet database (https://www.disgenet.org/, accessed on 30 June 2022) by selecting the diabetes-specific associations. Second, the T2D Knowledge portal (T2DKP) [20] is a meta-analytic summary of genome-wide associations from worldwide populations. We compared the significant associations found in this study with those reported in the T2D Knowledge portal. Third, we also searched the Harmonize database [21] for diabetes-specific gene associations in the context of those observed in the present study. Finally, we constructed a functional gene network of the significantly associated genes using the NetworkAnalyst web resource [22].

## 3. Results

### 3.1. Study Participants, Pools and the GSA Variants

The DISFIN study enrolled a total of 1444 participants whose detailed description has been given elsewhere [2]. From this group of participants, blood samples and all laboratory assays were available on 1402 (97%) participants and were included in the present study. The distribution of sociodemographic and clinical features of the included study participants (*n* = 1402) was similar to that in the entire study sample of 1444 participants. The prevalence of T2D, central obesity, hypertension and dyslipidemia in the study sample was 29.82%, 71.98%, 52.93% and 31.03%, respectively.

Whole blood pools were designed based on the presence of four metabolic conditions (T2D, central obesity, hypertension and dyslipidemia). From a potential 16 combinations of these metabolic conditions, we combined infrequent combinations (<1%, highlighted rows in Table 1) giving a total of 14 pools. Further, since these pools were used for genetic analyses in duplicates, we had a total of 28 study pools. The frequency of the 28 constructed whole blood pools is shown in Table 1. From a total of 665,608 variants included in the GSA platform, we included a total of 608,550 variants after excluding them based on sex chromosomes, mitochondria and those with a minor allele frequency (MAF) < 1%. The inclusion criteria for the variants and their distribution across the chromosomes are shown in Figure 1.

### 3.2. Pooled Blood and Genotyping Quality Control

The average GenTrain score (a measure of the single nucleotide polymorphism (SNP) calling quality reported by Illumina arrays) across the included variants was 0.8312 (95% confidence interval 0.8309–0.8314). The average estimate of the pooling error was −0.0067 (95% confidence interval was −0.0295–0.0162). Thus, both the genotyping and pooling error estimates were in the acceptable range.

### 3.3. Pooled GWAS Results—Variants

The Manhattan plot (Figure 2A) shows the results of our pooled GWAS study. From the total variants included in this study, 140 were found to be significantly associated with the risk of T2D at the genome level (*p* < 8.2163 × 10^−8^). The detailed identification and annotation of the significantly associated variants is provided in Appendix A. Briefly, significant variants were found on all chromosomes, but the majority were located on autosomes 3 and 7 (*n* = 13 each), 2 (*n* = 11) and 1 (*n* = 10). No strand preference was observed (69 and 71 variants on the + and—strand, respectively). The SNP calling quality of the significantly associated variants was comparable to that of the whole genome (average GenTrain score 0.8264). The average MAF was 25.42% and ranged between 1.01% to 49.86%. A total of 79 variants (56.43%) were within genes with known biological functions. The remaining 61 variants were located within an average of 73,632 bp (95% confidence interval 49,669–97,596 bp) of a gene with known biological function. The Q-Q plot (Figure 2B) showed that there was no systematic inflation (λ = 0.9517) at the level of the genome—indeed the Q-Q plot remained slightly below the line of expectation for all the variants except those that were significantly associated with T2D.

The top five most significantly associated variants are highlighted in Figure 2A,C. These were: rs1001179 (T→C polymorphism, P = 9.06 × 10^−25^), rs480948 (A→G polymorphism, P = 2.36 × 10^−24^), rs360745 (T→C polymorphism, P = 2.98 × 10^−24^)), rs7711236 (T→G polymorphism, P = 3.87 × 10^−24^) and rs73219073 (T→C polymorphism, P = 4.62 × 10^−24^). These variants were located on chromosomes 11, 11, 3, 5 and 21, respectively. The top first, second, third and fifth variants were related to the catalase (CAT), mastermind like transcriptional coactivator 2 (MAML2), IQ motif and Sec7 domain ArfGEF 1 (IQSEC1) and PR/SET domain 15 (PRDM15) genes, respectively. Violin plots showing the distribution of these variants in study pools with regard to the presence or absence of T2D are shown in Figure 2C.

### 3.4. Pooled GWAS Results—Genes

The significantly associated variants (*n* = 140) mapped to a total of 131 nearing or overlapping genes (Figure 3) of which 29 genes represent uncharacterized genes with no official names. We interrogated the relevance of the remaining 102 genes in the context of T2D by searching whether these genes are mentioned in the T2DKP (*n* = 4344 diabetes-related genes), DisGeNET (*n* = 2359 diabetes-related genes) or Harmonizome (*n* = 3381 diabetes-related genes) databases. We observed that nine of the significantly associated genes (ADCY5, MIP, EBF2, MITF, SLC30A8, NCAM1, HDAC9, CSF1 and DNASE1) were found in both T2DKP and in DisGeNET; 13 genes (IQSEC1, LHX2, SOX7, HSD17B12, PKNOX2, IGSF21, DNAH1, RPTOR, SAXO1, MAGI2, HNRNPAB, B4GALNT4 and ZNF385D) were found in T2DKP only while eight genes (CAT, ERG, CORO2B, ALMS1, TNFRSF11A, RYR2, KAZN and DPYS) were found in DisGeNET only. Further, the Harmonizome database identified another 22 genes (ARHGAP42, CA10, CACNA2D3, CGNL1, CLIC5, DLGAP1, FAT3, FRMD4A, GPR6, HECW1, IQCJ-SCHIP1, MAML2, MBOAT1, NCALD, NRP2, PBRM1, PTPRM, SLC24A3, SORCS2, SPOCK1, SYNDIG1 and TMEM132B) as diabetes-related from those found to be significant in this study. Thus, our study identified a total of 50 genes that were not included in T2DKP, DisGeNET or Harmonizome databases as diabetes related.

The detailed literature search revealed that 18 of these 50 genes have been associated with T2D previously even if these genes are not included in the T2DKP, DisGeNET and Harmonizome datasets. Published evidence implicated these 18 genes directly through population studies (GBP6, GSTO1, HHIPL1, INTS10, RGS16 and RNU6-679P), indirectly through association with other metabolic conditions (CCDC69, DNAH2, GALNT17, HERC5, KIF5C, MARCH2 and PRDM14) or through association with complications of T2D (CASC15, CCDC107, MIR147A, PFKFB3 and RNF166). Detailed annotation of the published literature in this regard is provided in Appendix A) [23,24,25,26,27,28,29,30,31,32,33,34,35,36,37,38,39,40,41]. From the remaining genes identified to be significantly associated with T2D in this study, there were two open reading frames, three long intergenic non-coding elements and a pseudogene. Thus, we identified a set of 22 genes as novel associations with T2D in the Sindhi population studied (Figure 3). These genes were: ARMH4, CDADC1, CYP2T3P, DRAIC, GNPTG, GUSBP6, HSPA8P13, MYOZ2, NDUFB9P2, PLEKHG1, PRDM15, RN7SKP144, RN7SKP203, RN7SKP250, RNF121, RNU4ATAC7P, RNVU1-6, RPL34P19, SIT1, TMTC1, TXNL1P1 and ZNF765.

### 3.5. Functional Relevance of the Significantly Associated Genes

Gene functional network analyses (Figure 4) showed that seven distinct functional blocks could be identified from the significantly associated genes. These blocks represented genes associated with the neuronal system functioning (CACNA2D3, CHRNA7, NCALD, ADCY5, RYR2, SLC24A3, SLC 30A8 and MIP); genes associated with Notch signaling (MAML2 and HDAC9); one gene each representing the diabetes-breast cancer nexus (BRCA2); vitamin C metabolism (GSTO1); mTOR signaling (RPTOR); pyrimidine catabolism (DPYS) and endothelial function (NRP2).

## 4. Discussion

Type 2 diabetes is a complex metabolic disease with a wide web of causation. It is now well known that T2D has a strong genetic basis, but the list of genes implicated in the web of T2D causation is not fully established. This is partly because there is a variation in the strength and specificity of the association (especially in the context of GWAS) of genomic variants across world populations. For example, most of the GWAS studies published relating to T2D have been conducted in the population of European and Asian ancestry with very few studies in African populations. A recent meta-analysis by Chen et al. [42] highlights this point. In this context, it is noteworthy that there are very few GWAS studies on T2D based on Indian populations. Tabassum et al. [43] published a GWAS largely based on the Indo–European and Dravidian population with the strongest association signal around the TMEM163 gene. In the same year, Saxena et al. [44] published a study on the Punjabi Sikh population with a novel signal at the SGCG gene. Recently, a study [45] published on the genetic basis of T2D in three ethnically endogamous Indian populations based in Tamil Nadu, Rajasthan and Andhra Pradesh states (the INDIGENIUS Consortium) also demonstrated substantial population-specificity in the genetic basis of T2D within Indian populations. It is therefore a need to conduct T2D GWAS studies in ethnically coherent subgroups within India. Our study represents an attempt in that direction with a focus on the Indian urban Sindhi ethnicity.

In this pooled whole blood GWAS study, we observed that after adjusting for comorbidities, genetic relatedness, pooling variation and batch effects, a total of 102 well-characterized genes were associated with T2D in the ethnically endogamous Sindhi population. Of these 102 genes, 70 genes had evidence of association consistent with previously published studies and there were 22 novel genes significantly associated with T2D that may indicate a Sindhi population-specificity. The functional relevance and generalization of these associations needs to be investigated in future studies. On the other hand, the functional relevance of the 70 well-characterized genes (shown in Figure 4) extracted some interesting functional blocks. First, the genes associated with the neuronal system were consistent with the fact the brain signaling system is commonly involved before and during the pathogenesis of T2D [46,47,48]. Notable among these are the ADCY5, SLC30A8 and SLC24A3 genes that are involved in neurohormonal transport and solute (especially sodium, calcium and zinc) transport. In addition, the gene coding type 2 ryanodine receptor (RYR2) regulates calcium release in the endoplasmic reticulum of the pancreatic beta cells and is thus known to be involved in insulin secretion and glycemic homeostasis [49]. Second, the association of the BRCA2 with T2D raises interesting possibilities in this population since simple dietary measures have been shown to significantly improve glucose metabolism in BRCA mutation carriers [50]. Third, the association of the RPTOR gene with T2D is also interesting. This gene partakes in the mTORC1/Raptor signaling pathway and is known to regulate beta-cell maturation and insulin synthesis [51]. Fourth, the role of Notch signaling is essential to beta-call maturation during the embryonic stage and during adult life [52,53]. We found two genes (MAML2 and HDAC9) that are strategically placed in the Notch signaling pathway and were associated with T2D in this study. Lastly, we also found association of genes implying vitamin C metabolism, pyrimidine metabolism and endothelial function regulation all of which make interesting targets for investigation in future studies.

It is also instructive to consider the potential mechanistic contribution of the top five most significantly associated variants (SAV, shown in Figure 2C). The topmost significantly associated variant (rs1001179) maps to the regulatory moiety of the CAT gene that codes for the catalase protein. This variant influences the transcription factor binding and thereby regulates catalase expression [54]. Concordantly, it has been demonstrated that lower circulating concentration of catalase is significantly associated with the risk of T2D [54,55]. The second most SAV (rs480948) mapped on to the MAML2 gene for which substantial evidence exists supporting its participation in the Notch signaling pathway as described above. Musicant et al. [56] have demonstrated that genomic rearrangements involving the MAML2 gene are associated with an inhibitory influence on the PPARγ and IGF-1 expression. Similarly, the fifth most SAV (rs73219073 related to the PRDM15 gene) is also critically placed in the genetics of T2D. This gene has been shown to partake in the regulation of PI3K/KRT/mTOR pathway and during the embryonic pancreatic development [57]. Thus, for three of the five SAV, there exists substantial published evidence for potential involvement in the pathophysiology and genetics of T2D. To our knowledge, such supporting evidence is not available for the remaining two SAVs (rs360745 and rs7711236).

There are strengths and weakness in our study. Among strengths, first this is a pooled GWAS that identified most of the known gene–diabetes associations with a few novel associations. The fact that ~70% of the associations observed in this GWAS have been previously reported by other scientific groups in other populations lends an indirect credence to the observations. These observations also have a high plausibility considering the strong biological rationale that supports the association studies. Second, due to the whole pooling blood approach used in the study, we needed 28 blood pools compared to the sample size of 1402 individuals—a genotyping cost reduction by 98% with acceptable pooling error. McGregor et al. [19] have previously shown that this approach is effective in picking up association signals accurately. Arguably, pool construction error may be less important than genotyping error due to the arrays [58]. In resource-limited settings, such an approach can greatly facilitate the first genetic screens for many other conditions. Future studies can then specifically address the candidate genes identified during this screening. Third, our study sample used a family-based approach. While most of the significantly associated variants had MAF > 10%, the family-based approach implicitly places confidence in the observed associations of variants with lower MAFs.

Conversely, our study has some limitations. First, the pooled GWAS approach is both a strength and a weakness. Presumably, a regular GWAS on all individuals (*n* = 1402) would have led to similar inferences since the estimated pooling error was low and acceptable. However, in the absence a full GWAS study, it is not possible to comment about the robustness of the observed associations. Second, due to the ethnically endogamous and genetically related nature of the study population, it is practically very challenging to design another such cohort for validation purposes. As a result, generalization of the observed associations is neither possible nor desired. We only aimed to find the association signals in the light of those reported from other populations within and outside India. Third, biological explanation for the functional role of the genes is currently not available and cannot be inferred from this study. For an efficient resource utilization, future studies need to specifically investigate the functional role of the genes through transcriptomic studies. We do not have gene expression data on the candidate genes of interest identified by our study, but future studies need to conduct gene expression analyses. We therefore compared the associations observed in this study with those reported in well-established repositories and in other published studies.

## 5. Conclusions

Without over-interpretation of the data, we conclude that our pooled whole blood GWAS in families of T2D patients uncovered interesting candidate genes for future investigation. While 70% of the observed associations have been previously identified and investigated by other research groups in different populations, we also report 22 novel gene-T2D associations that need to be interrogated more fully in future studies. To that end, our study also points towards the possibility of population-specificity of the observed associations in the urban Sindhi population in India studied here.

## Figures and Tables

**Figure 1 genes-13-01298-f001:**
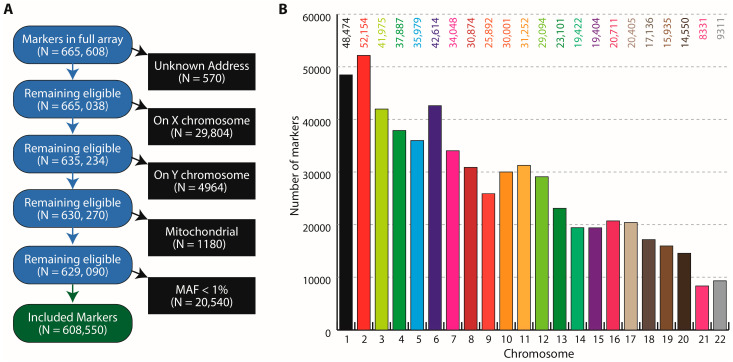
Genome-wide single nucleotide polymorphism (SNP) markers used in the pooled GWAS study. (**A**) Inclusion criteria for markers. Excluded markers are shown in black boxes, candidate markers are shown in blue boxes and the final number of included markers are shown in the green box. (**B**) Distribution of the included SNP markers across the genome. Bars show the number of markers on the indicated chromosome. The numbers are shown at the top of each bar.

**Figure 2 genes-13-01298-f002:**
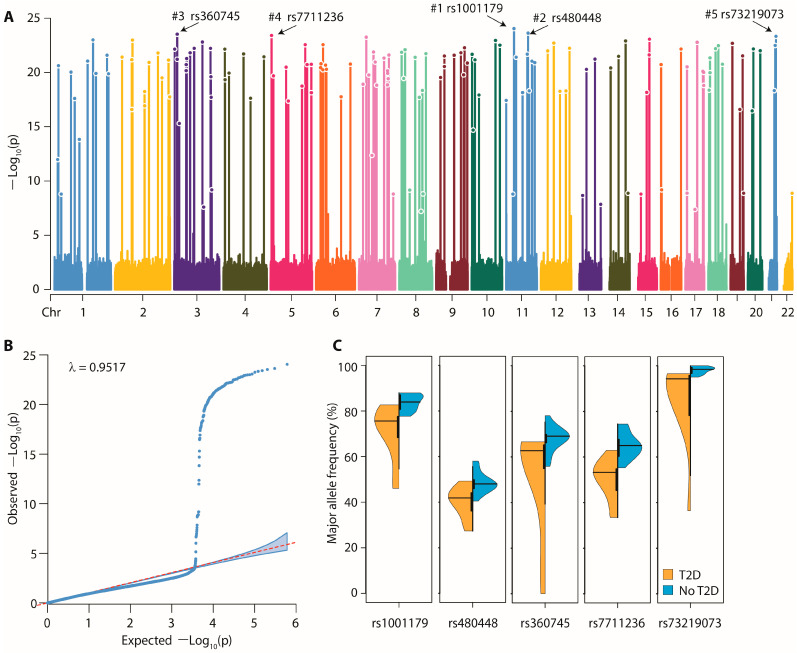
Results of the whole blood pooled genome-wide association study. (**A**) Manhattan plot. The points and droplines indicate the log-transformed, adjusted and statistically significant *p*-values. The five topmost significant associations are numbered as 1 through 5 and the corresponding SNP markers are shown at the top of the plot. (**B**) QQ plot. The plot shows the relationship between observed and expected *p*-value distribution (with 95% confidence bands). The genomic inflation factor (λ) is shown at the top of the plot. (**C**) Violin plots for the association of the top five significant SNP markers.

**Figure 3 genes-13-01298-f003:**
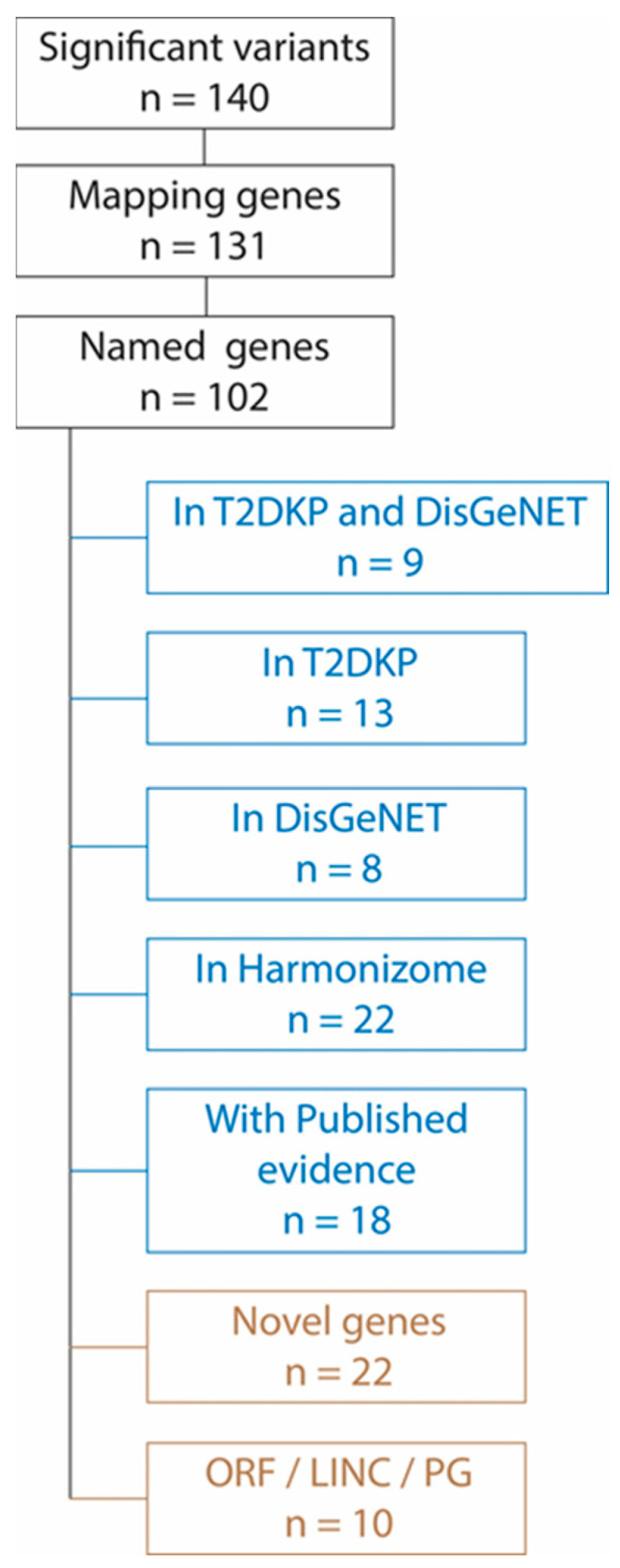
Characterization of the genes associated with T2D. T2DKP, Type 2 Diabetes Knowledge Portal; Dis-GeNET, the Disease–Gene Network database; ORF, open reading frame; LINC, long intergenic non-coding element; PG, pseudogene.

**Figure 4 genes-13-01298-f004:**
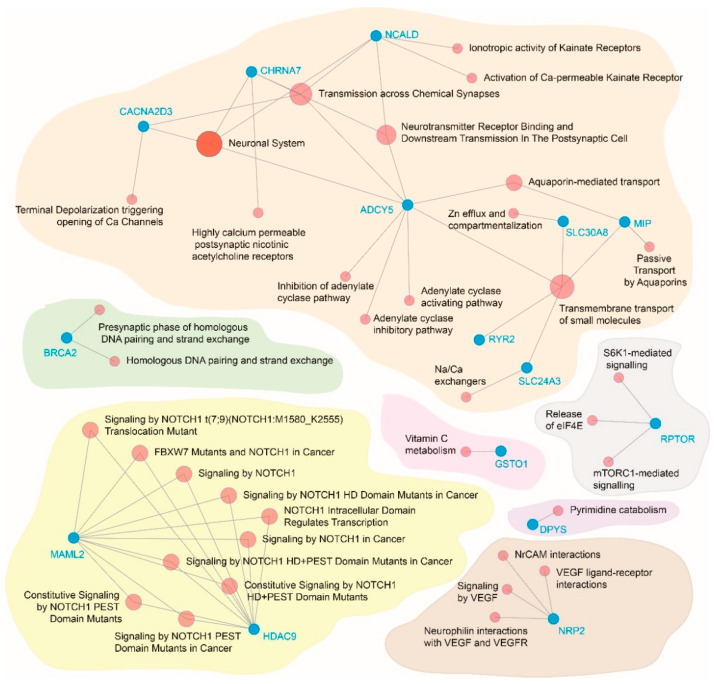
Functional gene network identified from the genes significantly associated with type 2 diabetes. Color coded irregularly shaped polygons represent the functional blocks. Blue nodes represent genes while pink nodes represent functional ontogeny.

**Table 1 genes-13-01298-t001:** Description of whole blood pools.

Pool Id	T2D	CO	HTN	DL	*n*	%
1	No	No	No	No	190	13.55
2	No	No	No	Yes	49	3.50
3	No	No	Yes	No	65	4.64
4	No	No	Yes	Yes	30	2.14
5	No	Yes	No	No	230	16.41
6	No	Yes	No	Yes	90	6.42
7	No	Yes	Yes	No	224	15.98
8	No	Yes	Yes	Yes	106	7.56
9	Yes	No	No	No	22	1.57
	Yes	No	No	Yes	7	0.50
10	Yes	No	Yes	No	19	1.36
	Yes	No	Yes	Yes	11	0.78
11	Yes	Yes	No	No	38	2.71
12	Yes	Yes	No	Yes	34	2.43
13	Yes	Yes	Yes	No	179	12.77
14	Yes	Yes	Yes	Yes	108	7.70

T2D, type 2 diabetes; CO, central obesity; HTN, hypertension; DL, dyslipidemia.

## Data Availability

Lata Medical Research Foundation’s Institutional Ethics Committee (LMRF-IEC) does not allow public data sharing to avoid potential identification. If data is requested for verification of results, we will seek permission from the LMRF-IEC before the requested data can be released. For further clarification of the LMRF-IEC’s data access policy as well as for data access requests, please contact: Dr. Prabir Kumar Das, Member Secretary, Institutional Ethics Committee, Lata Medical Research Foundation, Kinkine Kutir, Vasant Nagar, Nagpur—440022, Ph. No. 91-8805023450, Email: prabir_das23@rediffmail.com.

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
