# Peer review of "Susceptibility Loci for Type 2 Diabetes in the Ethnically Endogamous Indian Sindhi Population: A Pooled Blood Genome-Wide Association Study"

_genes, 2022, doi:10.3390/genes13081298_

Round 1

Reviewer 1 Report

The authors have conducted this study very well with supportive tables and figures. However, the authors need to revise the manuscript based on the comments given.

 If the authors present the results with the numerical values [percentages, CI, p-values], will be much better in the abstract section.

Why the authors have mentioned that Sindhi ethnic group in India are encountered the challenges of stress and migration. This needs an explanation with supporting evidence.

The introduction is well covered with details by covering the previous research conducted by the authors.

During which period the samples were collected? Hope it can be mentioned in the methodology.

The authors could add up the information about the 5 variants of the respective genes that are associated with the other ethnicities/populations, particularly among Asians will be more informative to know the significant association of T2D. If no such studies show significant association then authors could discuss that too.

Gene expression studies on those variants? Any supportive evidence?

How the research outcomes will be helpful in the clinical settings?

Table S1. Characteristics of the study participants are almost similar to table 1 from Mamtani et al., 2021, doi: 10.1371/journal.pone.0257390. Hence, this needs to be excluded. It is understood the current study is the continuation of the previous study conducted by the authors.

Acronyms are not consistent especially for type 2 diabetes [T2D]. Many acronyms are not fully written for the first time; like HTN, GWAS…

Please mention the references cited for supplemental table 4 in the text too.

Author Response

Responses to Comments by Reviewer 1

Comment 1

“The authors have conducted this study very well with supportive tables and figures. However, the authors need to revise the manuscript based on the comments given.”

Our response

We sincerely thank the Reviewer for the excellent peer review.

Comment 2

“If the authors present the results with the numerical values [percentages, CI, p-values], will be much better in the abstract section.”

Our response

Thanks for the suggestion. We have now edited the Abstract according to the Reviewer’s suggestion.

Comment 3

“Why the authors have mentioned that Sindhi ethnic group in India are encountered the challenges of stress and migration. This needs an explanation with supporting evidence.”

Our response

We have added the materials to the Introduction section as suggested by the Reviewer (page 1, lines 40-43)

Comment 4

“The introduction is well covered with details by covering the previous research conducted by the authors.”

Our response

Thank you.

Comment 5

“During which period the samples were collected? Hope it can be mentioned in the methodology.”

Our response

We have added the requested information to the Methods section (page 2 lines 76-77).

Comment 6

“The authors could add up the information about the 5 variants of the respective genes that are associated with the other ethnicities/populations, particularly among Asians will be more informative to know the significant association of T2D. If no such studies show significant association then authors could discuss that too.”

Our response

Thank you for the suggestion. We have now added the information as suggested by the Reviewer (page 8, lines 356-369; page 9, lines 370-372).

Comment 7

“Gene expression studies on those variants? Any supportive evidence?”

Our response

Again, we are highly indebted to the Reviewer for this suggestion. We have added the suggested information to the Discussion section (page 8, lines 356-369; page 9, lines 370-372).

Comment 8

“How the research outcomes will be helpful in the clinical settings?”

Our response

We think that the research into genetic basis of T2D in the Sindhi ethnic population is not yet fully understood. Our study provides the first clues in this direction but will need larger populations and robust validation across populations before the clinical impact of these findings can be gauged.

Comment 9

“Table S1. Characteristics of the study participants are almost similar to table 1 from Mamtani et al., 2021, doi: 10.1371/journal.pone.0257390. Hence, this needs to be excluded. It is understood the current study is the continuation of the previous study conducted by the authors.”

Our response

While the structure of this Table is similar to the Table published in the PLoS One paper, the contents are different and were shown with an aim to demonstrate that the minority participants who were not included in this study did not systematically differ from the majority who were included. However, as the Reviewer correctly points out the contents are very similar to the table published in the PLoS One paper. Hence as suggested by the Reviewer, we have now removed this table from the supplement.

Comment 10

“Acronyms are not consistent especially for type 2 diabetes [T2D]. Many acronyms are not fully written for the first time; like HTN, GWAS.”

Our response

We apologize for this oversight and have corrected it in the revised manuscript.

Comment 11

“Please mention the references cited for supplemental table 4 in the text too”.

Our response

Thanks. Done.

Reviewer 2 Report

This manuscript analyzed the type 2 diabetes gene by a pooled blood genome-wide association study, the study is very interesting and provides new ideas for the treatment of type 2 diabetes, but there are some minor issues, as follows:

1 The results of this article would be more reliable if the gene expression associated with type 2 diabetes could be validated using molecular biology techniques.

2 Whether there is a regulatory relationship between individual genes of the type 2 diabetes-related genes screened in this article, whether they regulate existing type 2 diabetes-related genes and form a network

Author Response

Responses to Comments by Reviewer 2

Comment 1

“This manuscript analyzed the type 2 diabetes gene by a pooled blood genome-wide association study, the study is very interesting and provides new ideas for the treatment of type 2 diabetes, but there are some minor issues, as follows:”

Our response

Thank you for the constructive review.

Comment 2

“1 The results of this article would be more reliable if the gene expression associated with type 2 diabetes could be validated using molecular biology techniques.”

Our response

We are grateful for this valid suggestion by the Reviewer. At this time, we do not have gene expression data but we understand and recommend that future studies need to conduct gene expression analyses. We have added this point to the limitation section as implied by the Reviewer (page 10, lines 402-404).

Comment 3

“2 Whether there is a regulatory relationship between individual genes of the type 2 diabetes-related genes screened in this article, whether they regulate existing type 2 diabetes-related genes and form a network”

Our response

This is another excellent suggestion by the Reviewer and ties in with the Comment 6 by Reviewer 1. We have now added further published evidence existing to hypothesize the mechanistic role of genes identified in this study in the pathophysiology of T2D (page 8, lines 356-369; page 9, lines 370-372).